# The impact of subject positioning on body composition assessments by air displacement plethysmography evaluated in a heterogeneous sample

Raluca Horhat[1,2], Monica Miclos-Balica[1], Paul Muntean[1], Sandra Popa[1], Irina Sima[1], Bogdan Glisici[1], Onisim Cîrja[1], Adrian Neagu[1,2,3], Monica Neagu[1,2]*

1 Department of Functional Sciences, Victor Babeş University of Medicine and Pharmacy, Timişoara, Romania, 2 Center for Modeling Biological Systems and Data Analysis, Victor Babeş University of Medicine and Pharmacy, Timişoara, Romania, 3 Department of Physics and Astronomy, University of Missouri, Columbia, Missouri, United States of America

* neagu.monica@umft.ro

**Data Availability Statement:** All relevant data are within the paper and its Supporting Information files (S1 File).

## Abstract

### Introduction

This study sought to evaluate the impact of subject positioning on body composition assessments by air displacement plethysmography using the BOD POD®.

### Methods

Eighty-two adults (42 men and 40 women), aged 26.1 ± 8.4 y (mean ± standard deviation), body mass index = 23.6 ± 4.8 kg/m², were assessed by repeated measurements in two different positions: relaxed (legs apart, back away from the rear) and compact (legs together, arms near the body, back touching the rear). We relied on Bland-Altman analysis to quantify the agreement between results recorded in the two positions. Using body surface charts, we tested the hypothesis that posture-induced variability stems from differences in exposed skin area.

### Results

Switching from compact to relaxed position resulted in a bias of -197 mL for body volume, -1.53% for percent body fat, and 1.085 kg for fat-free mass. The body surface area in contact with air was larger in relaxed position by 3632 ± 522 cm². When body volume was expressed in terms of the actual area of exposed skin in the compact position, the percent body fat bias became 0.08%, with a 95% confidence interval of (-0.14, 0.29)%.

### Conclusions

Subject posture is a source of significant variability in air displacement plethysmography. The disagreement between results obtained in different positions can be eliminated by adjusting the surface area artifact, suggesting that subject positioning in the BOD POD®

**Funding:** The authors received no specific funding for this work.

**Competing interests:** The authors have declared that no competing interests exist.

should be controlled to avoid changes in the amount of air maintained under isothermal conditions by the body.

## Introduction

Air displacement plethysmography (ADP) is widely used to determine body volume and assess body composition in the framework of two-component or multi-component models [1]. ADP is a non-invasive, fast, and undemanding technique, commercially available under the trade name BOD POD® (COSMED, USA) [2]. One ADP test includes two or three body volume (*BV*) measurements, each taking less than one minute, during which the subject sits in a hermetically closed fiberglass chamber and breathes normally. Meanwhile, the air from the chamber is acted upon by a diaphragm that oscillates at a frequency of 3 Hz. Since the amplitude of volume perturbations (0.35 L) is much smaller than the test chamber's volume (450 L), the corresponding pressure fluctuations are barely perceivable by the subject – their amplitude is about 0.1% of the ambient pressure. The instrument's software computes the volume of air in the test chamber by analyzing the pressure oscillations; then, *BV* is obtained by subtracting the volume of air from the volume of the chamber [3].

The validity of ADP was first established against hydrostatic weighing, the gold standard method of body composition analysis by densitometry [2]. ADP has been validated in various subjects, including healthy adults [4–6], elderly people [7, 8], and children [6, 9]. Despite its excellent overall accuracy, ADP has a slight tendency to overestimate adiposity in women [4, 10] and underestimate adiposity in men [4] – both by about 1% body fat. The reliability of ADP has also been a subject of interest for a number of studies [2, 11, 12], and it was found to be higher for inanimate objects than for human subjects [11].

The difficulty in measuring the volume of a human body resides in the thermodynamic complexity of the problem. The air maintained at constant temperature by the body is softer (i.e. more compressible) than the rest of the air from the test chamber, which is unable to exchange heat during the relatively short compression/decompression cycles [2, 3]. As the subject sits in the test chamber, the air in the lungs is kept under isothermal conditions, as well as is a thin layer of air in contact with the skin, whose volume is proportional to the body surface area (*BSA*). Hair and clothing also influence the amount of isothermal air next to the body surface [10, 13–15]. Therefore, the manufacturer recommends tight-fitting minimal clothing and hair be completely covered by a swim cap, with no air pockets underneath. Airtight silicone swim caps are advisable, especially for women with long hair, because they enable a more thorough elimination of the isothermal air trapped between hairs [16].

If the subject breathes regularly, the average thoracic gas volume (*TGV*) is the volume of air in the lungs at mid-exhalation. It can be measured using the BOD POD [2] or predicted by the instrument's software based on gender, age, and height [17].

The raw body volume (*BVr*) is defined as the body volume derived under the assumption that the subject is an inanimate object, and, thus, all the air in the test chamber is under adiabatic conditions [3]. That is, *BVr* is the volume of the object that could replace the subject in the test chamber and the instrument would record the same pressure oscillations as before. Taking into account the air maintained under isothermal conditions, *BV* is given by [1, 3]:

$$BV = BVr + |k| \cdot BSA + 0.4 \cdot TGV, \tag{1}$$

where $|k| \cdot BSA$ is the absolute value of the surface area artifact (*SAA*). The second term in

**Table 1. Subject characteristics, listed as mean ± SD as well as lower and upper bounds for men (M) and women (W).**

|  | Age (y) | Weight (kg) | Height (cm) | BMI (kg·m⁻²) |
|---|---|---|---|---|
| M (n = 42) | 26.5±7.7 (19.5, 48.2) | 81.19±16.63 (57.25, 130.10) | 178.8±6.6 (161, 190.2) | 25.3±4.9 (18.8, 41.5) |
| W (n = 40) | 25.7±9.1 (18.5, 49.2) | 57.59±11.14 (44.13, 97.54) | 162.5±5.7 (151.6, 175) | 21.8±3.9 (16.2, 33.8) |

Eq (1) is commonly written as $-SAA = -k \cdot BSA$. Since the constant of proportionality is a negative number, $k$ = -4.67·10⁻⁶ L/cm² [18], we chose to write Eq (1) in terms of its absolute value, $|k|$, showing that both sources of isothermal air bring positive contributions to $BV$.

A single ADP test includes at least two consecutive assessments of the $BV$. These are deemed consistent by the instrument's software if they differ by at most 150 mL; otherwise, the operator is instructed to perform a third measurement. This protocol assures an excellent precision of ADP measurements, with a technical error of measurement of about 1% $BF$ [12].

Possible implications of subject positioning in ADP have been investigated previously based on the hypothesis that the volume of the lungs is smaller in bent forward position than in straight position, thereby influencing $BV$ assessment [18]. However, the difference between the $TGV$s measured in the two positions was not significant and the author concluded that the difference between the $BV$s recorded in the two positions originated from the $SAA$. The overall compressibility of the air from the test chamber was smaller with the subject in straight position, leaning against the backrest, because less air was kept under isothermal conditions. Consequently, the amplitude of pressure oscillations was larger, and the software evaluated a smaller volume of air in the test chamber (i.e. larger $BV$) than in the case when the subject leaned forward. Consequently, the subject's adiposity was overestimated by 0.5% $BF$ in straight position compared to the bent forward position [18].

Despite its careful design and thorough statistical analysis, the study conducted by Peeters [18] has two limitations: (i) focusing on $TGV$, it involves subject positions that do not differ much from the point of view of the $SAA$, and (ii) it has been conducted on a homogeneous sample of 25 young men.

A recent study [19], involving 24 college-aged students, found mean differences as large as 1.42% $BF$ while investigating 4 postures – "seated normally, seated with an arched back, seated while leaning forward, and a maximal surface exposure condition in which subjects positioned their body to maximize airflow around them" [19].

The present work compares the results of ADP measurements performed in two positions that differ in $SAA$, while being in accord with the instructions of the BOD POD's manufacturer [17]. Moreover, it investigates a heterogeneous study group that is large enough to characterize the influence of posture on ADP test results for each sex. To show that the differences between $BV$ assessments performed in the two positions indeed result from the $SAA$, we used body surface charts [20] to evaluate the difference between the exposed $BSA$s in the two positions.

## Materials and methods

### Subjects

This study was performed on a gender-balanced, heterogeneous sample of 82 healthy adults (42 men and 40 women), recruited via flyers from University staff and students. Written informed consent was obtained from all participants. The study protocol was approved by the Committee of Research Ethics of our institution.

Table 1 presents the descriptive statistics of the study population in terms of mean ± standard deviation (SD) and the range of values (min., max.).

## ADP measurements

Body composition assessments were performed using the BOD POD® Gold Standard Body Composition Tracking System (COSMED, USA).

At the beginning of each day of measurements, a complete quality check procedure, including scale calibration, was conducted on the plethysmograph.

To prepare for ADP tests, subjects were asked to abstain from alcohol consumption for two days, avoid intense exercise for 12 hours, and to refrain from drinking or eating for 4 hours. Subjects were instructed to use the bathroom right before testing. First, standing height was measured to the nearest 0.1 cm using a wall-mounted tape measure (GIMA 27335, GIMA, Gessate, Italy) while the subject was barefoot and held her/his Frankfort plane horizontally. Body mass was determined to the nearest 0.001 kg, using the scale connected to the BOD POD. During an ADP test, the subject wore dry, minimal, tight, form-fitting clothing (lightweight swimsuit or spandex shorts and a non-padded sports bra). Hair was completely covered by a swim cap and no air pockets were left below the cap. The predicted estimate for the *TGV* was used and body composition was determined using Siri's equation [21]. The evaluated body composition parameters were *BVr*, *BV*, *%BF* and fat-free mass (*FFM*).

The subject's body composition was assessed in each of two positions: (i) *relaxed*, with legs apart, arms distanced from the thorax and the back not touching the backrest (Fig 1A and 1B), and (ii) *compact*, with legs together, arms near the body and the back in close contact with the rear (Fig 1C and 1D). Both of them are compatible with the BOD POD's manual, which instructs the technician to measure *BV* "while the subject sits comfortably in the test chamber" [17].

The repeated measures protocol proposed by Tucker et al. [22] was carried out in each position. One assessment consisted of at least two complete trials conducted in a row. If they were at most 1% *BF* apart, we took the mean of the results obtained in the two trials; otherwise, a third trial was performed and the result was given as the mean of the closest pair. All the measurements involving a given participant were performed on the same day, successively, within less than 40 minutes. The subject exited the measurement chamber after every trial.

## Correction of the surface area artifact

In the compact position, less skin is left in contact with the surrounding air than in the relaxed one. To evaluate the difference in exposed *BSA* between the two positions, we used body surface charts designed to assist burn patient care. For the results reported in the main paper, we used a modified Lund-Browder chart (S1 Fig) [23], which is stratified according to the nutritional status of the subject – quantified in terms of the body mass index (BMI), defined as body mass (kg) divided by height squared (m$^2$). The Supporting Information presents further results obtained with (i) the original Lund-Browder chart [24], (ii) the "Rule of Nines" [25] (S2 Fig), and (iii) a revised version of the "Rule of Nines", which takes into account the BMI [26] (S3 Fig).

When a person adopts the compact position, the lower limbs are in close contact with each other, the arms are close to the thorax, the forearms are close to the abdomen and the back is in contact with the backrest (Fig 1C and 1D). As a rough estimate of the hidden portions of the skin, we assumed that the back is in close contact with the rear, and about 1/6 of the area of each limb is covered by the opposite limb, thorax, or abdomen.

According to the modified Lund-Browder chart [23] (S1 Fig), when a person switches from the relaxed position to the compact one, the exposed skin area becomes smaller by:

$$\left( 2 \cdot \left( \frac{1}{6} \cdot 6.25 \cdot 2 + \frac{1}{6} \cdot 3.5 \cdot 2 + \frac{1}{6} \cdot 3 \cdot 2 \right) + 17.5 \right) \% \cdot BSA = 26.0\% \cdot BSA \qquad (2a)$$

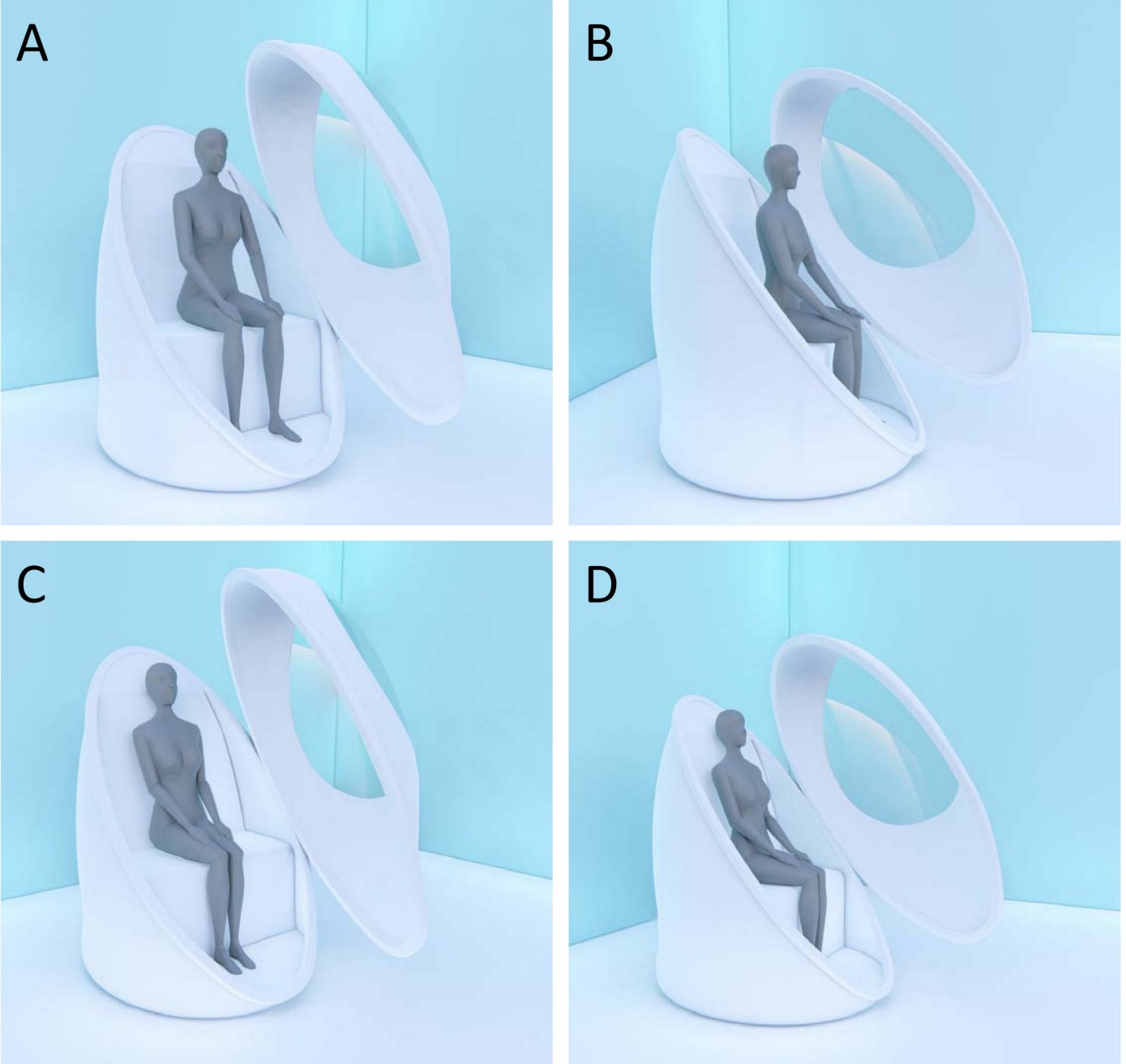

**Fig 1. Illustration of subject positioning.** Frontal view (A) and side view (B) of the subject in relaxed position; frontal view (C) and side view (D) of the subject in compact position.

for all but obese subjects (i.e. for persons with BMI $< 30 \text{ kg/m}^2$),

$$\left(2 \cdot \left(\frac{1}{6} \cdot 5.5 \cdot 2 + \frac{1}{6} \cdot 3 \cdot 2 + \frac{1}{6} \cdot 3 \cdot 2\right) + 20\right)\% \cdot BSA = 27.7\% \cdot BSA \qquad (2b)$$

for subjects with 30 kg/m$^2 \le$ BMI $\le$ 39.9 kg/m$^2$, and

$$\left( 2 \cdot \left( \frac{1}{6} \cdot 4.75 \cdot 2 + \frac{1}{6} \cdot 2.5 \cdot 2 + \frac{1}{6} \cdot 3 \cdot 2 \right) + 22.5 \right) \% \cdot BSA = 29.3\% \cdot BSA \qquad (2c)$$

for morbidly obese subjects (BMI $\ge$ 40 kg/m$^2$).

Hence, in the compact position, the corrected body volume was expressed as follows:

$$BV^{\text{corrected}}_{\text{compact}} = BVr + 0.740 \cdot |k| \cdot BSA + 0.4 \cdot TGV \qquad (3a)$$

if BMI $<$ 30 kg/m$^2$

$$BV^{\text{corrected}}_{\text{compact}} = BVr + 0.723 \cdot |k| \cdot BSA + 0.4 \cdot TGV \qquad (3b)$$

if 30 kg/m$^2 \le$ BMI $\le$ 39.9 kg/m$^2$

$$BV^{\text{corrected}}_{\text{compact}} = BVr + 0.707 \cdot |k| \cdot BSA + 0.4 \cdot TGV \qquad (3c)$$

if BMI $\ge$ 40 kg/m$^2$.

The corresponding value of the % *BF* was computed using Siri's formula [21].

## Statistical analysis

The results of this study are reported as mean ± standard deviation (SD). Normality of the data was established using the Shapiro-Wilks test. Statistical significance was defined as p<0.05. We performed a paired samples t-test or the Wilcoxon test to analyze differences between body composition parameters determined in the two positions. Prior to the study initiation, we estimated the number of subjects necessary to correctly reject the null hypothesis – the statement that subject posture does not influence ADP results. Calculations performed for a two-tailed test, with a power of 0.8, assuming posture-induced differences of 58 mL in raw body volume reported in the literature [18] gave a minimum of 38 subjects. Bland-Altman plots [27] were used to evaluate the agreement between the results of ADP tests performed in different positions. Statistical analysis of the recorded data was carried out in the R programming language [28].

## Results

Normal distribution was found for %*BF* values (p = 0.2 for relaxed position and p = 0.14 for compact position). For the other body composition parameters, the p-value of the Shapiro-Wilks test was below 0.001. Therefore, we applied the paired samples t-test to assess the statistical significance of the mean difference between %*BF* values obtained in the two positions, whereas for the other parameters we used the Wilcoxon test. When the subject repositioned from compact to relaxed state, statistically significant changes were observed in *BVr*, *BV*, %*BF*, and *FFM*, as demonstrated by the small p-values of the t-test (Table 2, row 3) or Wilcoxon test (Table 2, rows 1, 2 and 4).

Bland-Altman plots of differences vs. means of quantities recorded in relaxed and compact positions indicate a negative bias for *BVr* and *BV*, as well as a significant trend, with larger differences for greater volumes (Fig 2A and 2B, respectively). Hence, when the subject is in compact position, the *BVr* is overestimated by 193 mL in comparison to the value recorded in relaxed position. The same is true for *BV*, given by Eq (1), albeit to a slightly different extent (197 mL), due to different body mass values recorded in successive trials. (Note that *BSA* is calculated during each test based on the body mass recorded in that particular test.) For %*BF*, the bias was -1.53% *BF*, but no significant trend was observed (Fig 2C). Fat-free mass was underestimated by

**Table 2. Raw body volume (*BVr*), body volume (*BV*), percent body fat (%*BF*) and fat-free mass (*FFM*) assessed during repeated ADP trials with subjects in two positions (mean± standard deviation (SD)).**

|  | Relaxed | Compact | Relaxed-Compact | p |
|---|---|---|---|---|
| *BVr* (L) | 64.399±18.072 | 64.592±18.108 | -0.193±0.139 | <0.001 |
| *BV* (L) | 66.65±18.388 | 66.847±18.427 | -0.197±0.133 | <0.001 |
| *%BF* (%) | 22.66±9.59 | 24.19±9.84 | -1.53±0.97 | <0.001 |
| *FFM* (kg) | 53.349±13.268 | 52.264±12.995 | 1.085±0.695 | <0.001 |

1.09 kg in the compact state compared to the relaxed state and the linear regression of the differences vs. means was found to be statistically significant (p<0.001) (Fig 2D).

The body surface area in contact with air was larger in relaxed position by $3632 \pm 522$ cm$^2$ than in compact position. We evaluated the hypothesis that subject positioning affects ADP results because it influences the exposed *BSA*, and, thereby, the amount of air maintained under isothermal conditions by the body. To this end, we derived corrected formulas to express *BV* in terms of *BVr* for subjects seated in compact position (Eq (3)), and found excellent agreement with the results obtained in relaxed position via the standard procedure (Eq (1)), with a bias of 23 mL and no significant trend (Fig 3A). Percent body fat derived from the corrected *BV* given by Eq (3) displayed a bias of 0.08% *BF* (Fig 3B), whereas *FFM* had a bias of -4g (Fig 3C). When the *SSA* was adjusted for the compact position using Eq (3), no significant linear trend was observed in the differences vs. the means of data inferred in the two positions (Fig 3).

Bland-Altman plots, represented for each sex, before and after *SAA* adjustment are shown in S7–S11 Figs for women, and S12–S15 Figs for men. The corresponding parameters are listed in Table 3. For both sexes, the compact position yields higher % *BF* estimates than the relaxed one, but the effect of repositioning seems to be larger in men than in women – for males, the absolute value of the bias is 1.3 times larger than for females. The discrepancy between sexes is even more important when it comes to the assessed *FFM* – the corresponding bias is 1.76 fold larger for men than for women. Interestingly, the two sexes also differed in the trends observed in the Bland-Altman plots: for women, the differences between % *BF* values recorded in the two positions were larger for subjects with high adiposity, whereas for men an opposite trend was observed (compare S7 and S12 Figs). Hence, in our sample, the overestimation of adiposity in the compact position was similar (of about 1.5% *BF*) for both sexes in subjects whose % *BF* was higher than the sex-specific median.

The Supporting Information presents results derived from different body surface charts used in the care of burn patients [23–26] (S1–S3 Figs). The corresponding Bland-Altman plots are represented in S4–S6 Figs for the entire sample, in S8–S11 Figs for women, and S13–S15 Figs for men.

## Discussion

The results from this study are in accord with the pioneering work by Peeters [18], who observed that switching the subject from bent forward to straight position caused a 58 mL increment in the measured *BVr*, which could be assigned to a change in the *SAA*. Although statistically significant, the mean difference in *BV* observed by Peeters in young men was less than 150 mL, the maximum difference between two consecutive *BV* assessments considered consistent with each other [17]. The present work considers two positions that differ markedly in the fraction of *BSA* exposed to the surrounding air, and evaluates the impact of repositioning on ADP results for both sexes in a heterogeneous sample. In our study, the difference in *BVr* recorded in the two positions was $193 \pm 139$ mL, which corresponds to a difference of $1.53 \pm 0.97\%$ *BF*.

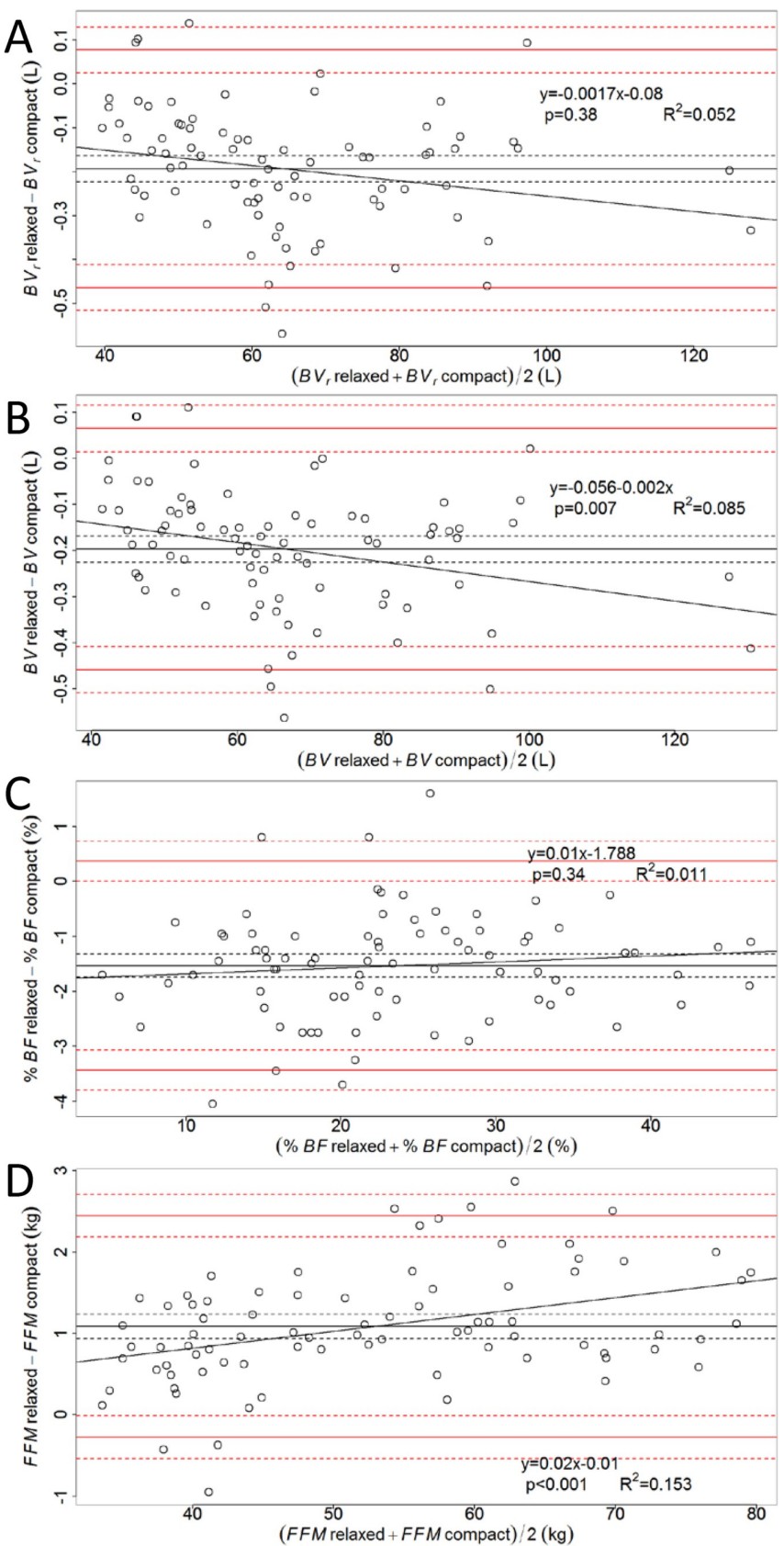

**Fig 2. Bland-Altman plots for *BVr, BV, %BF* and *FFM*.** Shown are the plots for *BVr* (A), *BV* (B), *%BF* (C), and *FFM* (D). In each plot, differences between results obtained in relaxed position and compact position are plotted vs. their mean; the black solid line represents the bias (the mean of the differences), whereas the black dashed lines delimit the 95% confidence interval (CI) of the bias; red solid lines depict the 95% limits of agreement, given by bias ± 1.96 times the standard deviation of the differences; red dashed lines show the corresponding 95% CI; linear regression equations of differences vs. means are given together with the p-value and the coefficient of determination ($R^2$).

Perle-Jones et al. [19] observed a mean difference of 0.97% *BF* between the normal posture and the leaning forward posture and 1.42% *BF* between arched back and leaning forward. Although it is unclear how the postures considered by Perle-Jones and colleagues [19] relate to ours, it seems reasonable to assume that the leaning forward posture presents similarities with the relaxed position (Fig 1A and 1B), whereas the other postures involve partially hidden skin surfaces; therefore, due to altered *SAA*, they lead to higher values of the measured adiposity.

For young men, Peeters [18] concluded that subject positioning only had a marginal effect on the results of BOD POD measurements. By contrast, in the present study subject repositioning inflicted changes in body composition parameters larger than the technical error of measurement

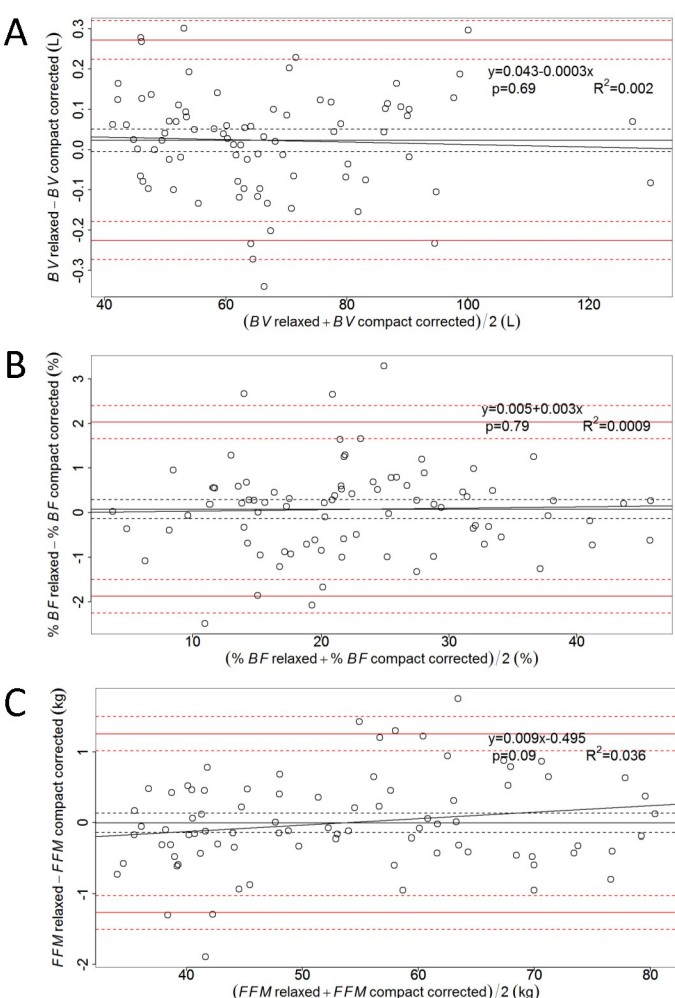

**Fig 3. Bland-Altman plots for *BV, %BF*, and *FFM* derived from the corrected *SAA* corresponding to the compact position.** Shown are plots of differences vs. means of *BV* (A), *%BF* (B), and *FFM* (C) obtained in relaxed position and compact position – with *BV* given by Eq (3).

**Table 3. Results of the Bland-Altman analysis of body composition parameters derived in relaxed position and compact position (relaxed—compact), before and after the surface area artifact (SAA) correction performed using Eq (3).** Listed are the bias and the upper limit of agreement (ULA) along with their 95% confidence intervals (CI).

| | | Before SAA corr. Bias (95% CI) | ULA (95% CI) | After SAA corr. Bias (95% CI) | ULA (95% CI) |
|---|---|---|---|---|---|
| All (n = 82) | BVr (L) | -0.193 (-0.223, -0.163) | 0.077 (0.025, 0.129) | - | - |
| | BV (L) | -0.197 (-0.226, -0.168) | 0.064 (0.014, 0.115) | 0.023 (-0.004, 0.050) | 0.271 (0.224, 0.319) |
| | %BF (%) | -1.53 (-1.74, -1.32) | 0.37 (0.002, 0.73) | 0.076 (-0.137, 0.291) | 2.03 (1.65, 2.40) |
| | FFM (kg) | 1.085 (0.930, 1.23) | 2.44 (2.19, 2.70) | -0.004 (-0.143, 0.135) | 1.259 (1.02, 1.50) |
| Men (n = 42) | BVr (L) | -0.239 (-0.282, -0.196) | 0.039 (-0.036, 0.115) | - | - |
| | BV (L) | -0.245 (-0.287, -0.204) | 0.021 (-0.051, 0.094) | -0.002 (-0.044, 0.040) | 0.27 (0.196, 0.344) |
| | %BF (%) | -1.73 (-2.00, -1.46) | 0.03 (-0.44, 0.51) | -0.21 (-0.47, 0.05) | 1.49 (1.03, 1.95) |
| | FFM (kg) | 1.375 (1.172, 1.577) | 2.69 (2.331, 3.045) | 0.167 (-0.036, 0.370) | 1.485 (1.127, 1.843) |
| Women (n = 40) | BVr (L) | -0.145 (-0.181, -0.109) | 0.084 (0.02, 0.148) | - | - |
| | BV (L) | -0.146 (-0.18, -0.111) | 0.072 (0.011, 0.132) | 0.049 (0.015, 0.082) | 0.261 (0.203, 0.321) |
| | %BF (%) | -1.33 (-1.64, -1.02) | 0.65 (0.10, 1.20) | 0.38 (0.05, 0.70) | 2.42 (1.85, 2.99) |
| | FFM (kg) | 0.781 (0.599, 0.964) | 1.934 (1.613, 2.256) | -0.185 (-0.36, -0.008) | 0.929 (0.618, 1.239) |

(TEM) of individual ADP tests. Indeed, the TEM of %BF assessments using the BOD POD was found in the range 0.55–1.28% BF: 0.55% BF in a sample of 21 young men [18], 0.57% BF in a pool of college-aged subjects (31 men and 31 women) [29], 0.8% BF in a sample of 102 adults (45 men and 57 women) [11], 1.07% BF in a large, heterogeneous sample of healthy adults (548 men and 432 women) [12], and 1.28% BF in a sample of 8 men and 16 women [30].

The Bland-Altman plot of Fig 2A indicates a significant trend between the difference of BV readings performed in the two positions and their mean, with larger differences corresponding to larger mean BVs. A similar trend was observed also by Peeters [18], albeit the bias was 2.3 times smaller in absolute value than in the present study.

To identify the cause of the variability induced by subject positioning, we evaluated the hypothesis that repositioning affects the measured BVr because of the change in the volume of air maintained in isothermal state in close vicinity to the body. Taking into account the difference in exposed BSA, we derived formulas that furnished roughly the same BV in the compact position as the value obtained in relaxed position, thereby confirming our working hypothesis.

The Lund and Browder chart [24], and the "Rule of Nines" formulated by Wallace and Pulaski [25] are widely used in the clinics, but they often lead to an overestimation of the burn extent, especially in obese patients [31]. Both the Lund and Browder chart and the "Rule of Nines" have been revisited by taking into account the nutritional status of the patient: Neaman et al. proposed a modified Lund-Browder chart [23], whereas Mance et al. found that the "Rule of Nines" was accurate for men, but needed a BMI-dependent stratification for women [26]. The present study relies on correction formulas derived from the modified Lund-Browder chart (S1 Fig), the original Lund-Browder chart (S2 Fig, panel B), the original "Rule of Nines" (S2 Fig, panel A), and the revisited "Rule of Nines" (S3 Fig). Using different charts employed in burn patient care [23–26] (S1–S3 Figs), we obtained slightly different values for the bias, but their order of magnitude was the same. Comparing the Bland-Altman plots of Fig 3 with those of S4–S6 Figs, we notice that the correction formula given in Eq (3) provides the best overall performance (the smallest absolute values for the bias and for the slope of the linear regression). Nevertheless, the "Rule of Nines" furnished a surprisingly good correction of the BV recorded in compact position.

It is important to stress, however, that Eq (3) is not meant to be used instead of Eq (1)–the one that is implemented in the BOD POD's software; Eq (3) has been devised merely to identify the mechanism responsible for the influence of subject positioning on ADP results.

It is unlikely for the extreme positions to pass unnoticed during one complete ADP trial, since they often lead to differences in BV larger than the acceptable threshold of 150 mL; the software would instruct the technician to perform one more measurement because of the inconsistency of the first two readings [17]. Nevertheless, it is worth noticing that, although the BV bias for the two extreme positions (197 mL) was larger than the consistency threshold, in 33.6% of the subjects the difference in BV between the two positions was lower than 150 mL.

The limitations of this work include the relatively small sample size and the rough approximations implied in the estimation of the area of hidden skin in the compact position. Our sample was heterogeneous and large enough to study each sex in part, but it did not allow for further stratification of the results according to age or nutritional status. The use of body surface charts provided a reasonable estimate of the SAA. Nevertheless, the assumption that in compact position each limb is covered in a proportion of 1/6 is simplistic. Depending on body constitution, various portions of a limb might be covered to different extents. Three-dimensional body scanners could provide more precise anthropometric information [32, 33], but mobile phone apps are also promising tools for personalized estimates of the exposed body surface area [20].

In conclusion, this study aimed to evaluate the impact of subject positioning on ADP assessments of body composition using the BOD POD. To this end, we carried out repeated measurements with the subject in two extreme positions, albeit in accord with the instrument's operation manual [17]. In the relaxed position, the subject had much of her/his BSA in contact with the surrounding air, whereas in the compact position about 1/4 of the BSA was hidden (covered by the backrest or by nearby body parts), leading to an underestimation of the volume of air left in the test chamber (i.e. an overestimation of BV). The effect of repositioning was larger for men than for women. We found that a proportional reduction of the SAA in the expression of the measured BV precisely compensates this effect, yielding similar body composition parameters as in relaxed position. Therefore, it seems safe to conclude that the cause of the variability induced by subject positioning is the change in the amount of air maintained under isothermal conditions in the proximity of the body.

This study reinforces the message of previous works [18, 19], that there is a need for a standardized protocol of subject positioning during body composition assessments by ADP. Posture control is especially important while tracking the outcome of a dietary and/or lifestyle intervention. In a longitudinal study, consecutive assessments are made weeks or months apart, so the subject is likely to sit in various positions unless instructed otherwise. The relaxed position considered in this work seems to be optimal because it is comfortable and maximizes the fraction of body surface in contact with the air from the test chamber. Further research will be needed to evaluate the accuracy and precision of ADP tests performed in a standard posture.

## Supporting information

**S1 Fig. The modified Lund-Browder chart that takes into account nutritional status [23].** On these schematic drawings of the front (left) and back (right) of the human body, the number displayed on, or next to, each body part gives its area as a percentage of the body surface area. Certain body parts are labeled by letters and their areas are listed below the schemes as a function of BMI. (The drawings were reproduced and modified with permission from the work of Cheah et al. [20]).
(PDF)

**S2 Fig.** The most common body surface charts used for evaluating burn patients: (A) the "Rule of Nines" [25], and (B) the Lund-Browder chart [24]. In each panel, shown are various body parts, viewed from the front (left scheme) and back (right scheme). The numbers

displayed on, or next to, body parts express their areas as percentages of the body surface area. (The schematic drawings from this figure were reproduced with permission from the work of Cheah et al. [20]).
(PDF)

**S3 Fig. The "Rule of Nines" for women, revised by taking into account their BMI [26].** The drawings represent the front and back of the body (left and right, respectively); letters label body parts whose surface areas are listed below the drawings, as percentages of the body surface area.
(PDF)

**S4 Fig. Bland-Altman plots for *BV*, *%BF*, and *FFM*, obtained by using the Lund-Browder chart [24] to estimate the actual *SAA* associated to the compact position.** The plots represent differences vs. means of *BV* (A), *%BF* (B), and *FFM* (C) obtained in relaxed position and compact position (upon correction). In each panel, the black solid line represents the bias, defined as the mean of the differences, whereas the black dashed lines delimit the 95% confidence interval (CI) of the bias. Red solid lines represent the 95% limits of agreement (the lower and upper limit of agreement, given by bias $\pm 1.96 \times$ the standard deviation of the differences), whereas red dashed lines depict the corresponding 95% CI. Also, the equation of the regression line, the corresponding p-value, and the coefficient of determination ($R^2$) are displayed on each plot.
(PDF)

**S5 Fig. Bland-Altman plots for *BV*, *%BF*, and *FFM*, derived from the modified "Rule of Nines" [26] assessment of the actual *SAA* associated to the compact position.** The plots represent differences vs. means of *BV* (A), *%BF* (B), and *FFM* (C) obtained in relaxed position and compact position (upon correction). Notations are explained in the caption of S4 Fig.
(PDF)

**S6 Fig. Bland-Altman plots for *BV*, *%BF*, and *FFM*, generated by applying the "Rule of Nines" [25] to estimate the actual *SAA* corresponding to the compact position.** The plots represent differences vs. means of *BV* (A), *%BF* (B), and *FFM* (C) obtained in relaxed position and compact position (upon correction). Notations are explained in the caption of S4 Fig.
(PDF)

**S7 Fig. Bland-Altman plots of *BVr*, *BV*, *%BF*, and *FFM* of female subjects.** The plots represent differences (relaxed-compact) vs. means of *BVr* (A), *BV* (B), *%BF* (C), and *FFM* (D). Notations are explained in the caption of S4 Fig.
(PDF)

**S8 Fig. Bland-Altman plots for *BV*, *%BF*, and *FFM*, obtained for women by using the modified Lund-Browder chart [23] to estimate the actual *SAA* associated to the compact position.** The plots represent differences vs. means of *BV* (A), *%BF* (B), and *FFM* (C) recorded in relaxed position and compact position (after correction). Notations are explained in the caption of S4 Fig.
(PDF)

**S9 Fig. Bland-Altman plots for *BV*, *%BF*, and *FFM*, generated for women by applying the Lund-Browder chart [24] to compute the true *SAA* associated to the compact position.** The plots represent differences vs. means of *BV* (A), *%BF* (B), and *FFM* (C) measured in relaxed position and compact position (upon correction). Notations are explained in the caption of S4 Fig.
(PDF)

**S10 Fig. Bland-Altman plots for *BV*, *%BF*, and *FFM*, derived for women from the modified "Rule of Nines" [26] calculation of the true *SAA* corresponding to the compact position.** Plotted are differences vs. means of *BV* (A), *%BF* (B), and *FFM* (C) measured in relaxed position and compact position (after correction). Notations are explained in the caption of S4 Fig.
(PDF)

**S11 Fig. Bland-Altman plots for *BV*, *%BF*, and *FFM*, obtained for women by applying the "Rule of Nines" [25] to compute the actual *SAA* associated to the compact position.** Plotted are differences vs. means of *BV* (A), *%BF* (B), and *FFM* (C) measured in relaxed position and compact position (after correction). Notations are explained in the caption of S4 Fig.
(PDF)

**S12 Fig. Bland-Altman plots of *BVr*, *BV*, *%BF*, and *FFM* of male subjects.** The plots represent differences (relaxed-compact) vs. means of *BVr* (A), *BV* (B), *%BF* (C), and *FFM* (D). Notations are explained in the caption of S4 Fig.
(PDF)

**S13 Fig. Bland-Altman plots for *BV*, *%BF*, and *FFM*, generated for men by using the modified Lund-Browder chart [23] to calculate the actual *SAA* associated to the compact position.** The plots represent differences vs. means of *BV* (A), *%BF* (B), and *FFM* (C) measured in relaxed position and compact position (upon correction). Notations are explained in the caption of S4 Fig.
(PDF)

**S14 Fig. Bland-Altman plots for *BV*, *%BF*, and *FFM*, obtained for men by using the Lund-Browder chart [24] to compute the true *SAA* associated to the compact position.** The plots show differences vs. means of *BV* (A), *%BF* (B), and *FFM* (C) recorded in relaxed position and compact position (after correction). Notations are explained in the caption of S4 Fig.
(PDF)

**S15 Fig. Bland-Altman plots for *BV*, *%BF*, and *FFM*, derived for men based on the "Rule of Nines" [25] to calculate the actual *SAA* associated to the compact position.** The plots show differences vs. means of *BV* (A), *%BF* (B), and *FFM* (C) recorded in relaxed position and compact position (upon correction). Notations are explained in the caption of S4 Fig.
(PDF)

**S1 File. Microsoft Excel workbook of anonymized experimental data.** This workbook is composed of two worksheets (one for each sex) containing selected columns of the data file saved by the BOD POD's software. The second column, ID1, contains the identification number of the subject (in the range 1001–1042 for men and 2001–2040 for women), whereas the third column, ID2, specifies the posture (1-relaxed, 2-compact). Four data sets are included for each subject–one pair for each posture, given by the repeat measures protocol of Tucker et al. [22]: the first two readings if they were at most 1% *BF* apart, or, otherwise, the closest pair of readings out of three trials. A body composition variable associated to a given posture was computed by taking the mean of the corresponding pair of readings.
(XLSX)

## Acknowledgments

We thank Frujina Neagu for building the digital model of a person seated in the BOD POD's test chamber (Fig 1).

## Author Contributions

**Conceptualization:** Adrian Neagu, Monica Neagu.

**Data curation:** Sandra Popa, Irina Sima, Onisim Cîrja.

**Formal analysis:** Monica Miclos-Balica, Paul Muntean, Bogdan Glisici.

**Investigation:** Monica Miclos-Balica, Paul Muntean, Sandra Popa, Irina Sima, Bogdan Glisici, Onisim Cîrja.

**Methodology:** Monica Neagu.

**Software:** Raluca Horhat, Adrian Neagu.

**Supervision:** Monica Neagu.

**Writing – original draft:** Raluca Horhat.

**Writing – review & editing:** Adrian Neagu, Monica Neagu.

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
