## [Decision Letter · Decision Letter 0]

8 Feb 2022

PONE-D-21-15286The impact of subject positioning on body composition assessments by air displacement plethysmography evaluated in a heterogeneous samplePLOS ONE

Dear Dr. Neagu,

Thank you for submitting your manuscript to PLOS ONE. After careful consideration, we feel that it has merit but does not fully meet PLOS ONE’s publication criteria as it currently stands. Therefore, we invite you to submit a revised version of the manuscript that addresses the points raised during the review process.

I would like to sincerely apologize for the delay you have incurred with your submission. It has been exceptionally difficult to secure reviewers to evaluate your study. We have now received two completed reviews; their comments are available below. The reviewers have raised constructive concerns about the study that need to be addressed in a revision. In particular, please pay attention to Reviewer#2 comments about the Bland and Altman SI plots.

Please revise the manuscript to address all the reviewer's comments in a point-by-point response in order to ensure it is meeting the journal's publication criteria. Please note that the revised manuscript will need to undergo further review, we thus cannot at this point anticipate the outcome of the evaluation process.

We look forward to receiving your revised manuscript.

Kind regards,

Miquel Vall-llosera Camps, Ph.D.

Academic Editor

PLOS ONE

Journal Requirements:

Reviewers' comments:

Reviewer's Responses to Questions

**Comments to the Author**

1. Is the manuscript technically sound, and do the data support the conclusions?

Reviewer #1: Yes

Reviewer #2: Yes

2. Has the statistical analysis been performed appropriately and rigorously? 

Reviewer #1: Yes

Reviewer #2: Yes

3. Have the authors made all data underlying the findings in their manuscript fully available?

Reviewer #1: Yes

Reviewer #2: Yes

4. Is the manuscript presented in an intelligible fashion and written in standard English?

Reviewer #1: No

Reviewer #2: No

5. Review Comments to the Author

Reviewer #1: Overall, this is a really good study and I adds to the body composition literature on subject positioning using the BodPod. I encourage the author to correct past tense as well as grammar issues.

Abstract:

Line 29: I’d just state how many were in the study Eight-two healthy adults (42 men and 40 women etc….) How is healthy defined?

Line: 31. How is “extreme” defined? I suggest just taking it out and saying “two different position were utilized.

Introduction:

Line 60: The accuracy of ADP was confirmed 61 for various categories of subjects, including healthy adults [4, 5, 6], elderly [7, 8], and children [6, 62 9]. I would change was confirmed to The accuracy has been validated in various subjects……

Line 108: Take out “has”

Line115-116: Research doesn’t prove it merely suggests. Reword this sentence please.

Methods:

Line 135: What about nicotine or caffeine or alcohol consumption?

Line 136: Why did the void before arrival? Usually you have subjects VOID right before testing.

Line 200: Change “were done” to were analyzed.

Discussion:

Bringing the reader back to the purpose would be good.

Change all ref to the actually author Jones-Smith et al..

Reviewer #2: Thank you for your in-depth investigation of factors influencing calculation of the SAA for adults undergoing ADP assessments! See the attached file for this reviewer's comments/questions/suggestions.

6. PLOS authors have the option to publish the peer review history of their article (what does this mean?). If published, this will include your full peer review and any attached files.

Reviewer #1: No

Reviewer #2: No

---

## [Author Response · Author response to Decision Letter 0]

17 Feb 2022

Response to Reviewers 

We thank the reviewers for the thorough analysis of our manuscript! We equally appreciate their positive comments, which encourage us, and their constructive criticism, which hopefully helped us to improve the paper. 

 The reviewers’ comments are listed in this letter point-by-point, placed between quotation marks. Each comment if followed by our answer. In the revised manuscript, the modified portions of the text are highlighted using the Track Changes function of Microsoft Word. Line numbers are also given wherever the new text appears on a different line in the revised manuscript than in the original one. 

Reviewer #1 

Comment 1: "Overall, this is a really good study and adds to the body composition literature on subject positioning using the BodPod. I encourage the author to correct past tense as well as grammar issues."

We thank Reviewer #1 for the appreciative comment and for the help with pinpointing English usage issues. 

Comment 2: Line 29: "I’d just state how many were in the study Eight-two healthy adults (42 men and 40 women etc….) How is healthy defined?"

 We revised the Abstract accordingly. 

Comment 3: Line: 31. "How is “extreme” defined? I suggest just taking it out and saying “two different position were utilized."

 Done. 

Comment 4: Line 60: "The accuracy of ADP was confirmed 61 for various categories of subjects, including healthy adults [4, 5, 6], elderly [7, 8], and children [6, 62 9]. I would change was confirmed to The accuracy has been validated in various subjects……"

 Done. 

Comment 5: Line 108: "Take out “has”"

 Done (line 109 in the revised manuscript).

Comment 6: Line115-116: "Research doesn’t prove it merely suggests. Reword this sentence please."

 Done (lines 116-117 in the revised manuscript). 

Comment 7: Line 135: "What about nicotine or caffeine or alcohol consumption?"

 Indeed, we asked them not to consume alcohol for two days prior to being tested, but we did not mention smoking and coffee consumption. We revised the text accordingly (line 136 in the revised manuscript). 

Comment 8: Line 136: "Why did the void before arrival? Usually you have subjects VOID right before testing."

 Actually, we asked them to visit the bathroom right before commencing the first test. Hopefully, the rewritten sentence is less confusing than the original (line 138 in the revised manuscript). 

Comment 9: Line 200: "Change “were done” to were analyzed."

 Done (line 202 in the revised manuscript). 

Comment 10: "Change all ref to the actually author Jones-Smith et al.."

 Done. 

Reviewer #2 

Comment 1: "Thank you for your in-depth investigation of factors influencing calculation of the SAA for adults undergoing ADP assessments! Your findings and insights provide great depth of understanding as to why body positioning within the BodPod chamber is critical – especially for serial assessments of body composition."

 We thank Reviewer #2 for this encouraging comment! 

Comment 2: "To conform to the conventions and grammar of the English language, this reviewer suggests enlisting services of a native English speaker to help clarify your otherwise well-written report. Suggestions for both grammar, clarity, and content are below. If the journal provided editing services, this list would be much shorter."

 We appreciate the constructive criticism and the proposed improvements of English grammar and style! 

Line 48 – "“compartment” should be “component”"

 Done. 

Line 67 – "by “softer” do you mean “less dense”? Please clarify"

 The revised sentence states explicitly that “softer” refers to “more compressible”. 

Line 70 - "change to read “as well as is a …”"

 Done. 

Line 72 - "change to read “hair be completely…”"

 Done (line 73 in the revised manuscript). 

Line 84 – "this reviewer did not see the definition of BSA and TGV prior to their use in the BV formula"

 The acronym of the body surface area (BSA) is defined in the revised text on line 71, whereas that of the thoracic gas volume (TGV) appears on line 77.

Line 113 – "change to read “positions that concern”"

 The corresponding sentence was revised to state explicitly that the investigated positions differed in the surface area artifact, while being in accord with the manufacturer’s instructions. 

Line 128 – "use “lower and upper bounds” instead of “range” in table title"

 Done. 

Line 125 – "move “, including scale calibration,” to follow “procedure in line 134"

 Done (line 135 in the revised manuscript). 

Line 138 – "indicate if participants were barefoot for standing height"

 Done (line 140 in the revised manuscript). 

Line 141 – "change to read “wore dry, minimal….”"

 Done (line 142 in the revised manuscript). 

Line 153 – "be sure the sequence of posture in Fig 1 matches specifics of the figure title."

 Yes, the caption of Figure 1 is correct: panels A and B represent the subject in relaxed position, whereas panels C and D depict the subject in compact position, in which a significant part of her/his body surface area is hidden from the surrounding air. 

Line 154 – "would be helpful to identify if participants exited the chamber between trials in Tucker’s study." 

 Yes. they did exit the measurement chamber after every trial (line 161 in the revised manuscript).

Line 158 – "even though you are most interested in SAA, indicated if TGV was estimated or measured? If estimated, mention that as a limitation."

 The TGV was predicted by the BOD POD’s software, as stated on line 144. 

Line 163 – "change to read “In the compact…”"

 Done (line 163 in the revised manuscript). 

Lines 185-189 – "is “VTG” supposed to represent “TGV”? The latter is a more common term, Be sure to define the term prior to its use." 

 We thank Reviewer #2 for noticing this typo! The correct acronym is defined before its first use (line 77). 

Line 193 – "insert a space before ±"

 Done. 

Line 255 – "references to SI figures no longer follows ascending numbering sequence. Reference to S Fig 3 occurs on line 170, but no reference to S figures 4 -6 found prior to reference to S7 figure. Reference to S44-S6 comes on line 274. See below for comments regarding clarity of the figure titles/legends."

 We chose to group the BA plots derived from the entire sample, and then, figures S7-S11 represent the BA analysis of data obtained for women, whereas figures S12-S15 refer to men. This option is explained in lines 252-253. 

Line 277 – "Is it not more appropriate to indicate that it is “The results from this study are in accord…”?"

 We revised the sentence accordingly (line 275 in the revised manuscript). 

Line 282 – "consider changing to read “positions that concern the fraction…”"

 Done (line 280 in the revised manuscript).

Line 293 – "consider using “Peeters and colleagues [18]” instead of “ref [18]” as the subject of your sentence"

 The revised sentence is, hopefully, more straightforward and legible (line 291 in the revised manuscript). 

Line 303 – "change “ref [13]” to the first author’s name followed by [13]."

 Done (line 300 in the revised manuscript). 

Line 307 – "change to read “in close vicinity to the”"

 Done (line 304 in the revised manuscript). 

Line 313 – "consider clarifying who/what is being referred to aby “Both of them”. It seems you are referring to techniques, but it is not clear as written."

 The revised sentence opening avoids the pronoun: “Both the Lund and Browder chart and the “Rule of Nines” have been revisited ...” (line 309 in the revised manuscript).

Line 316 – "by “This study” are you referring to “The present study” meaning your study? Not clear as written."

 Indeed, it is clearer to write “The present study” (line 312 in the revised manuscript). 

Comment 3: "This reviewer did not see a reference to S13 or S16."

 Supplementary figures S7-S11 show the BA plots obtained for women, whereas S12-S15 refer to men. These sex-specific plots are mentioned on line 253 and the corresponding BA parameters are listed in Table 3. 

Comment 4: "In reference list, check format for journal title for #19. It seems to be the only one that spells out the complete journal title."

 The revised reference contains the abbreviated journal title (Int J Exerc Sci). 

Comment 5: "My primary concern is with the figure titles/descriptions of the Bland and Altman SI plots using both red and black line colors. Some of the SI figures have two solid red lines with 95% LOA dashed lines, but no indication I found as to what they were indicating. While this suggestion may be perceived by the authors as adding redundancy, it would clarify some confusion for readers like me who have extensive experience with Bland & Altman plots. It would also help with the concept about each figure being able to stand alone."

 The revised caption of S4 Fig. provides a detailed presentation of the notations and the elements displayed on each BA plot (lines 473-478). Furthermore, to guide a reader interested in another supplementary figure, the captions of S5-S15 Figs. were supplemented with the sentence “Notations are explained in the caption of S4 Fig.”. 

Comment 6: "Another concern was the inability to understand “the numbers displayed …” aspect of the title for SI Fig. Providing an example would go a long way in providing clarity of what you are trying to say."

 The explanations given in the caption of S4 Fig. address this problem, too.

---

## [Decision Letter · Decision Letter 1]

4 Apr 2022

The impact of subject positioning on body composition assessments by air displacement plethysmography evaluated in a heterogeneous sample

PONE-D-21-15286R1

Dear Dr. Neagu,

We’re pleased to inform you that your manuscript has been judged scientifically suitable for publication and will be formally accepted for publication once it meets all outstanding technical requirements.

Kind regards,

Carla Pegoraro

Staff Editor

PLOS ONE

Additional Editor Comments (optional):

Reviewers' comments:

Reviewer's Responses to Questions

**Comments to the Author**

1. If the authors have adequately addressed your comments raised in a previous round of review and you feel that this manuscript is now acceptable for publication, you may indicate that here to bypass the “Comments to the Author” section, enter your conflict of interest statement in the “Confidential to Editor” section, and submit your "Accept" recommendation.

Reviewer #1: All comments have been addressed

Reviewer #2: All comments have been addressed

2. Is the manuscript technically sound, and do the data support the conclusions?

Reviewer #1: Yes

Reviewer #2: Yes

3. Has the statistical analysis been performed appropriately and rigorously? 

Reviewer #1: Yes

Reviewer #2: Yes

4. Have the authors made all data underlying the findings in their manuscript fully available?

Reviewer #1: Yes

Reviewer #2: Yes

5. Is the manuscript presented in an intelligible fashion and written in standard English?

Reviewer #1: Yes

Reviewer #2: Yes

6. Review Comments to the Author

Reviewer #1: (No Response)

Reviewer #2: Very fun to read again! Nice job on addressing grammatical issues and providing additional clarity. Please check line 243 and change "SSA" to "SAA" if it is indeed a typo. Good job, and congratulations!

7. PLOS authors have the option to publish the peer review history of their article (what does this mean?). If published, this will include your full peer review and any attached files.

Reviewer #1: No

Reviewer #2: No

---

## [Editor Report · Acceptance letter]

7 Apr 2022

PONE-D-21-15286R1 

The impact of subject positioning on body composition assessments by air displacement plethysmography evaluated in a heterogeneous sample 

Dear Dr. Neagu:

I'm pleased to inform you that your manuscript has been deemed suitable for publication in PLOS ONE. Congratulations! Your manuscript is now with our production department. 

Kind regards, 

on behalf of

Dr Carla Pegoraro 

Staff Editor

PLOS ONE